# 3D motion tracking display enabled by magneto-interactive electroluminescence

Seung Won Lee[1,4], Soyeon Baek[1,4], Sung-Won Park[2], Min Koo[1], Eui Hyuk Kim[1], Seokyeong Lee[1], Wookyeong Jin[1], Hansol Kang[1], Chanho Park[1], Gwangmook Kim[1], Heechang Shin[2], Wooyoung Shim [1], Sunggu Yang[3], Jong-Hyun Ahn [2] & Cheolmin Park [1✉]

Development of a human-interactive display enabling the simultaneous sensing, visualisation, and memorisation of a magnetic field remains a challenge. Here we report a skin-patchable magneto-interactive electroluminescent display, which is capable of sensing, visualising, and storing magnetic field information, thereby enabling 3D motion tracking. A magnetic field-dependent conductive gate is employed in an alternating current electroluminescent display, which is used to produce non-volatile and rewritable magnetic field-dependent display. By constructing mechanically flexible arrays of magneto-interactive displays, a spin-patchable and pixelated platform is realised. The magnetic field varying along the z-axis enables the 3D motion tracking (monitoring and memorisation) on 2D pixelated display. This 3D motion tracking display is successfully used as a non-destructive surgery-path guiding, wherein a pathway for a surgical robotic arm with a magnetic probe is visualised and recorded on a display patched on the abdominal skin of a rat, thereby helping the robotic arm to find an optimal pathway.

[1] Department of Materials Science and Engineering, Yonsei University, Seoul 120-749, Korea. [2] Department of Electrical and Electronic Engineering, Yonsei University, Seoul 120-749, Korea. [3] Department of Nano-Bioengineering, Incheon National University, Incheon 22012, Korea. [4]These authors contributed equally: Seung Won Lee, Soyeon Baek. ✉email: cmpark@yonsei.ac.kr

Human-interactive displays (HIDs) facilitate the visualisation of sensible and invisible information such as touch, smell, and sound and have attracted significant interest owing to their potential in emerging IoT-connected wearable electronics[1–6]. Owing to the tremendous demand for electronic skin (e-skin), which can artificially mimic the properties of human skin such as the ability to sense pressure, temperature, and humidity, most HIDs have emphasised on the visualisation of these functions, giving rise to a variety of human-interactive pressure[6–11], temperature[12], and humidity[13,14] displays. The visualisation of phenomena that is rarely sensible, such as magnetic fields, ultrasonic waves, and odourless toxic gases or liquids, can further broaden the utilisation of HIDs[15–19].

Particularly, the development of e-skins that are capable of sensing magnetic fields has been of great significance because they can imitate the magnetoreception that some animals and insects utilise for their orientation and navigation[20–24]. Furthermore, magneto-interactive e-skins can facilitate the detection of a magnetic field-sensitive object without physical contact, making them potentially suitable for three-dimensional (3D) touchless systems. From this viewpoint, numerous flexible, wearable, and skin-patchable magnetic sensors have been developed based on various physical principles, such as giant magnetoresistance[20,21,25], anisotropic magnetoresistance[22,26], spin valves[27], magneto-impedance[28], Hall effect[29], and magnetic composites[23,24,30]. Besides the efficient sensing of the magnetic field, the visualisation of the magnetic field has considerable technological potential through human extrasensory interactive displays. Moreover, when the magnetic field is detected and visualised simultaneously and is stored and retrieved several times after the removal of magnetic field, the magneto-interactive display with non-volatile memory function can be further utilised.

Herein, a skin-patchable 3D motion tracking display enabled by magneto-interactive electroluminescence (EL) is presented by simultaneously sensing, visualising, and memorising various magnetic fields. An alternating current (AC) EL display is developed with a magnetoactive conductive fluid of multi-walled carbon nanotubes (MWNTs), decorated with superparamagnetic iron oxide ($Fe_3O_4$) nanoparticles, in $n$-hexadecane on which an AC-field responsive emissive layer is placed with two parallel electrodes, as shown in Fig. 1a. When a magnetoactive conductive channel is formed upon an external magnetic field serving as a magnetic field gate, the display emits the characteristic of AC EL. With a decrease in the impedance of magnetoactive conducting channel with magnetic field, the luminance of the device increases, allowing for the facile visualisation of the magnetic field in EL. In addition, an EL is erased by destroying the channel by applying opposite magnetic field. Subsequently, writing a new EL with another input magnetic field is successfully performed, making our non-volatile magneto-interactive EL display (NV-MED) rewritable. By utilising the dependence of magnetic field on the distance along the z-axis between a magnetic probe and NV-MEDs, a 3D motion tracking display is realised with $5 \times 5$ pixelated NV-MED arrays. Our 3D motion tracking display, enabled by the magneto-interactive EL, was patched onto rat skin to efficiently visualise and memorise the 3D internal route of a surgery robot arm with a magnetic probe, thereby facilitating the search for an optimal pathway without the possibility of damaging the surgery spot.

## Results

**Fabrication of an NV-MED**. To fabricate NV-MED, an emissive layer of 40 μm-thick ZnS:Cu/poly(vinylidene fluoride-tri-fluoroethylene-chlorofluoroethylene) [PVDF-TrFE-CFE] composite was prepared by spin-coating on two in-plane transparent indium tin oxide (ITO) electrodes sputter-deposited onto a poly (ethylene terephthalate) (PET) substrate. Square-segmented VHB acrylic film was built on the emissive layer and used as spacer, which was capable of adjusting the amount of magnetoactive fluid. The magnetoactive fluid was carefully poured into the area defined by the spacer, followed by the sealing of fluid with a PET cover to protect from possible leakage and evaporation of the magnetoactive fluid (Supplementary Figs. 1 and 2). The cross-sectional view of an NV-MED was obtained after fully drying n-hexadecane, followed by obtaining a cross section of the device with a focused ion beam; the results are shown in Fig. 1b. Four stacked layers were visualised by scanning electron microscopy (SEM) of PET/ITO/[ZnS:Cu/PVDF-TrFE-CFE]/$Fe_3O_4$-MWNTs, combined with the energy dispersive X-ray mapping of the constituent atomic elements of the layers (Supplementary Fig. 3). An NV-MED fabricated in a $2.5 \times 2.5$ cm$^2$ area is mounted on a finger, as shown in Fig. 1c.

**Principle of magnetic sensing and memory**. The AC field applied between the two parallel electrodes has a little effect on the light emission of ZnS:Cu particles of an NV-MED owing to the in-plane electric field, as shown in the left scheme of Fig. 1d. When the external magnetic field is applied vertically on the top of NV-MED, the $Fe_3O_4$-MWNTs responding to the external magnetic field are dragged towards the source magnet, thereby increasing the networked conductive channel in $Fe_3O_4$-MWNTs right underneath the emissive layer, as shown in the right scheme of Fig. 1d (Supplementary Fig. 4). Since the networked conductive channel serves as a floating electrode, the in-plane AC field between the two parallel electrodes before the magnetic field is converted into vertical one (the right scheme of Fig. 1d)[3,4], as confirmed by an electrical field calculation based on finite element analysis, shown in Fig. 1e. The degree of percolation network of the $Fe_3O_4$-MWNTs, which depends upon the external magnetic field, significantly affects the total NV-MED impedance because the $Fe_3O_4$-MWNTs behave as a highly switchable conductor. Under these circumstances, the proposed device can detect the change in the impedance as a function of magnitude of external magnetic field. Furthermore, the vertical AC field dependent upon the impedance of the floating electrode of the networked $Fe_3O_4$-MWNTs can initiate the device, with the EL from the solid-state cathode luminescence of the ZnS:Cu particles in PVDF-TrFE-CFE varying in intensity as a function of magnetic field. In addition, a magnetic field visualised in EL is stored and retrieved several times, even after the magnetic field is removed, as described in detail in later sections.

**Characterisation of the magnetoactive fluid and its performance optimisation**. First, in the presence of an impedance change, we systematically investigated the sensing and memory performance of the magnetic field of NV-MED with a characterised magnetoactive fluid (Supplementary Figs. 5–8) and optimised magnetoactive channel of $Fe_3O_4$-MWNTs in n-hexadecane (Supplementary Figs. 9–11). We explicitly examined how the impedance of our NV-MED was varied with MWNTs having different diameters and lengths. In a given length of MWNTs, the sensitivity of impedance variation was rarely affected with the diameter of the nanotubes. As expected, the absolute impedance in a given magnetic field was much lower with long MWNTs than short ones due to facile formation of networked conductive channel with long nanotubes. However, the long MWNTs exhibited relatively low sensitivity of the impedance to the magnetic field, making them less suitable for our NV-MED (Supplementary Fig. 12). In our magneto-interactive display, a non-polar solvent is favourable because it

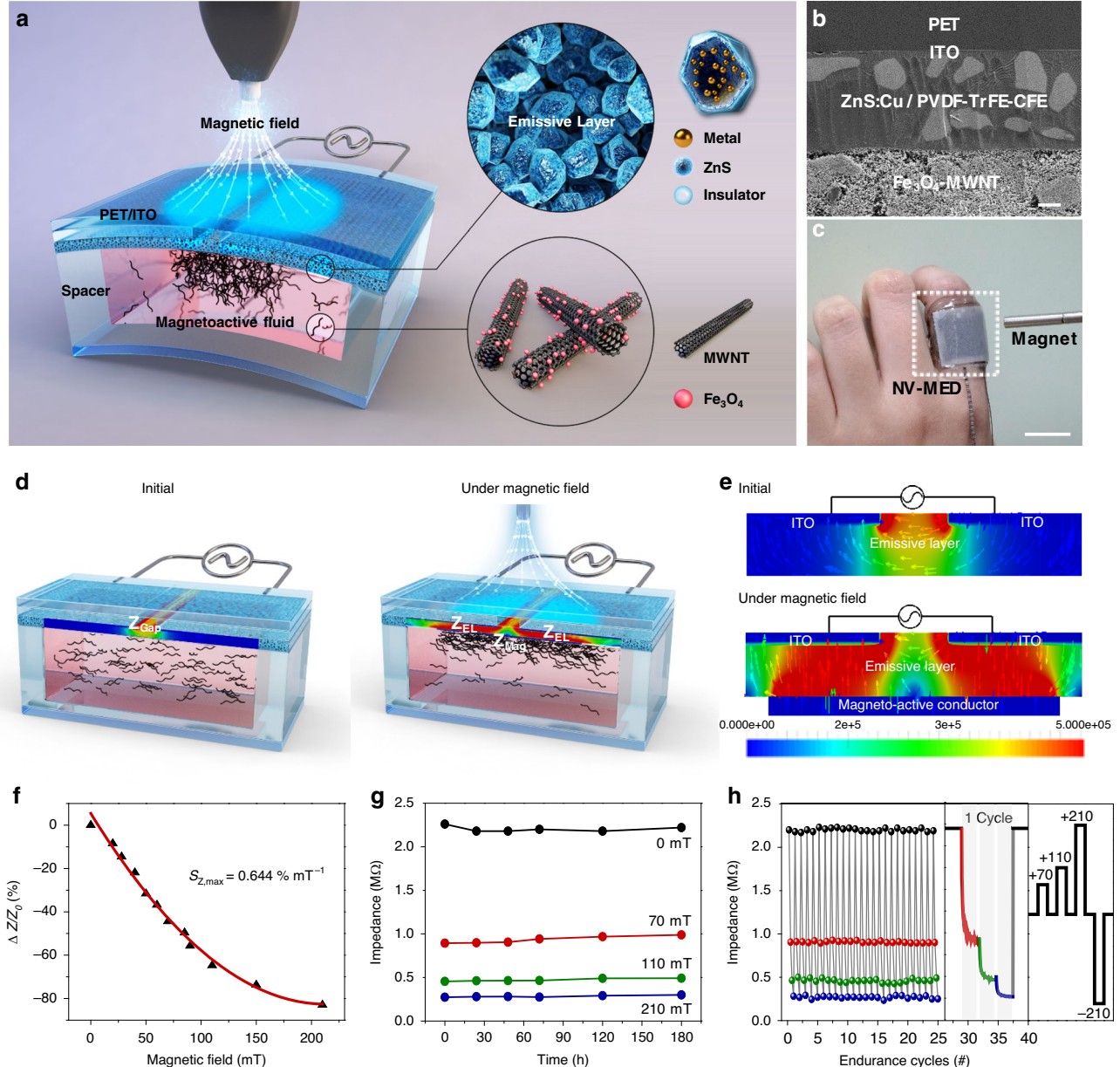

**Fig. 1 Device architecture and working principle of NV-MED. a** Conceptual illustration of an NV-MED containing magnetoactive fluid with two in-plane electrodes. A floating electrode is developed by the percolation network of magnetoactive fluid. **b** A cross-sectional SEM image of an NV-MED (scale bars: 1 μm). **c** NV-MED mounted on the back of a hand (scale bars: 1 cm). **d** Schematic of the electrical circuit of an NV-MED in the reading process. **e** FEA of AC field analysis of an NV-MED with distribution of electric field magnitude on parallel-type electrodes upon reading and erasing, respectively. **f** Plots of the change in impedance intensity of an NV-MED as a function of magnetic field. **g** Time-dependent retention of the impedance intensity arising from an NV-MED. **h** Write–erase cycle endurance in EL intensity changes of an NV-MED. Inset shows the EL intensity change in one cycle and scheme of four successive magnetic field programmes for the endurance cycle.

provides the large impedance variation before and after the development of a conductive bridge of $Fe_3O_4$-MWNTs, making the sensitivity of our device maximized. We chose n-hexadecane as a non-polar solvent in addition because of its dispersion capability of MWNTs with its appropriate viscosity which also affected device switching rate (Supplementary Fig. 13). In the aspect of biocompatibility of n-hexadecane, it is noted that according to the material safety data sheet, n-hexadecane can cause skin problems when spilled. To address the biocompatibility of a solvent, we employed several skin compatible solvents to our NV-MEDs (Supplementary Fig. 14). We found that both mineral oil and olive oil with the dielectric constants similar to that of n-hexadecane were also suitable for our device. Those

solvents are, however, more viscous than n-hexadecane, making the switching rate a littler lower (Supplementary Table. 1).

The impedance of NV-MED decreased when magnetic field up to 210 mT was applied. The impedance sensitivity of the device was evaluated as a function of magnetic field, as shown in Fig. 1f, defined as $S_Z = \delta(\Delta Z/Z_0)/\delta B$, where B is the applied magnetic field, and Z and $Z_0$ are the impedances with and without the applied magnetic field, respectively. A high sensitivity of 0.644% $mT^{-1}$ was obtained for a magnetic field below 110 mT, whereas the sensitivity decreased to ~0.166% $mT^{-1}$ for a magnetic field between 110 and 210 mT. NV-MED exhibited a fast impedance response, with its relaxation time following the square-shaped consecutive ON and OFF pattern of the magnetic field

(Supplementary Fig. 15). In this experiment, four distinct magnetic fields, i.e. 70, 110, 210vand −210 mT, were programmed in terms of different impedance values, and each impedance was preserved for a long time in the absence of a magnetic field, as shown in Fig. 1g. We also confirmed that the impedance levels of the four states were reliably developed upon 25 programme/erase cycles without significant variation. Impedance was rapidly decreased once a magnetic field was applied to an NV-MED. Further slight decrease in impedance occurred with fixation of the conductive channel during the magnetic pulse of 2 s. When the magnetic field was removed, the conductive MWNT channels with magnetic nanoparticles were slightly relaxed in viscous solvent medium. As a result, the conductivity of the channel slightly decreased, giving rise to the slight increase of impedance as shown in Fig. 1h.

**Sensing and memorising the magnetic field in EL**. Magnetic field applied on NV-MED was directly visualised and memorised in EL over a broad magnetic field range upon AC operation; the results are shown in Fig. 2. Because the AC field depends upon the network impedance, the light intensity of an NV-MED increases with magnetic field, as shown in Fig. 2a. The NV-MED emitted the characteristic blue light of the ZnS:Cu particles when various circular-shaped magnetic fields were applied, as shown in Fig. 2b. Notably, the light emission of the NV-MED under AC operation was observed near the edges of the two parallel ITO electrodes, even without a magnetic field, owing to the dielectrophoretic alignment of the $Fe_3O_4$-MWNTs arising from the parallel AC fringe field. The measured EL intensity arising from the dielectrophoretic force was three or four orders of magnitude lower than that resulting from the magnetophoretic one, making the dielectrophoretic contribution negligible in our NV-MED (Supplementary Figs. 16)[31–34]. The NV-MED EL changes are plotted as a function of magnetic field, as shown in Fig. 2c. In particular, under a magnetic field below 50 mT, the vertical AC field arising from networked $Fe_3O_4$-MWNT channel by the magnetic field was not sufficient to turn on EL in our device even though the device impedance was sufficiently lowered. The AC frequency significantly affected both the EL and impedance performance. The absolute impedance value decreases as the frequency increases since the impedance is a function of the frequency reciprocal. However, the sensitivity of impedance change to the magnetic field was rarely varied with frequency. We examined the EL characteristics and threshold magnetic field as a function of frequency (Supplementary Figs. 17). Although absolute luminescence was enhanced with increasing frequency of up to 10 kHz, the luminescence was rapidly dropped with further increase. Consistent with other AC EL device, the threshold magnetic field of ~50 mT was rarely altered with the frequency. The EL sensitivity is defined as $S_{EL} = \delta(\Delta L/L_0)/\delta B$, where B is the applied magnetic field, and L and $L_0$ are the EL intensities with and without the applied magnetic field at a certain AC field, respectively. A sensitivity of ~0.24898 mT$^{-1}$ was obtained for a magnetic field between 70 and 210 mT. The NV-MED exhibited a fast EL response, with its relaxation times following the square-shaped consecutive ON and OFF pattern of the magnetic field, as shown in Fig. 2d.

The proposed NV-MED was suitable for multi-state EL memory with which we were able to write, read, and erase various levels of magnetic fields in EL. First, we examined the time-dependent EL retention for four distinct programmed magnetic fields in the proposed NV-MED; and results are shown in Fig. 2e. All the luminance written with the magnetic-field levels of 70, 110 and 210 mT were well maintained for a duration longer than 180 h after removal of the magnetic field. In addition, we

examined the writing and erasing cycle endurance of the proposed NV-MED for four distinct impedance levels; the results are shown in Fig. 2f. The four different EL states were reliably developed upon 25 programme/erase cycles without significant variation. The EL-based magnetic field programme/erase process is shown using a series of NV-MED images in Fig. 2g. The direct visualisation of the magnetic field in EL with our NV-MED promoted the recognition of magnetic field shape, which was difficult to achieve with electronic devices based solely on the impedance, resistance, and capacitance. Both circular and square magnets are recognised in EL, as shown in Fig. 2h. We also confirm the endurance cycle test of EL intensity under a magnetic field of 210 mT, as shown in Fig. 2i.

**Patchable flexible pixelated NV-MED arrays**. Pixelated arrays of our NV-MEDs fabricated on a PET substrate was developed; results are shown in Fig. 3. We fabricated 5 × 5 arrays of pairs of in-plane parallel ITO electrodes (area of each set of pixel electrodes: $0.8 \times 0.8$ cm$^2$) with a 0.2 cm gap between each pair on a PET substrate, which can reduce the undesired edge-light emission by the fringe field (Supplementary Fig. 18). Most of EL devices based on AC are inevitable from high-voltage operation because they rely upon field induced light emission. Due to the high operation voltage of the AC EL devices, their safety should be carefully addressed for further development, in particular for skin patchable applications. To reduce the operation voltage of an NV-MED, we employed a high-k dielectric of PVDF-TrFE-CFE. Although the operation voltage of 150 V of our NV-MED is quite high, compared with conventional LEDs, it is significantly lower than that prepared with conventional dielectric elastomer[7,9,10]. It should be also noted that since light emitting phosphor particles are completely embedded in dielectric medium, the current during device operation is usually very low, making our device safely skin-patchable in spite of the rather high operation voltage (Supplementary Figs. 19). This was followed by the subsequent deposition of an emissive and magnetoactive layer (Supplementary Fig. 20). Individual pixels of the NV-MED arrays mounted on the skin can be written with a magnetic bar or pen, and the written information can be read in both the impedance and EL and erased many times, as schematically shown in Fig. 3a. For instance, the arrays of NV-MEDs were programmed with three different magnetic fields, i.e. $M_1$, $M_2$, and $M_3$ ($M_1 > M_2 > M_3$), to record the letter 'N', as shown in Fig. 3b. First, when the arrays operated with an AC voltage and frequency of ±150 V and 10 kHz, respectively, the programmed magnetic information on the arrays was clearly shown in the pixelated EL map, wherein three different impedance levels arising from the three input magnetic fields were apparent, as shown in Fig. 3c. The pixelated letter 'N' was simultaneously visualised with three different light intensities, as shown in Fig. 3d, 3e.

The pixelated NV-MEDs on a PET are mechanically flexible, making them suitable for wearable, skin-mountable applications. We investigated the bending impedance of a single-cell NV-MED; results are shown in Fig. 3f. An NV-MED of the size $2.5 \times 2.5$ cm$^2$ was programmed with a magnetic field of 210 mT and operated as a function of bending radius, with an AC voltage and frequency of ±150 V and 10 kHz, respectively. Both the EL intensity and impedance of the device were rarely altered at bending radii up to 5 mm. The rewritable performance of the arrays of our NV-MEDs was also examined by sequentially writing, reading, and erasing magnetic information, followed by the rewriting of new magnetic information. The letter 'N' programmed and visualised in EL in the arrays was completely erased and, subsequently, another letter, 'P', was successfully written and visualised. Then, another letter, 'L', was visualised

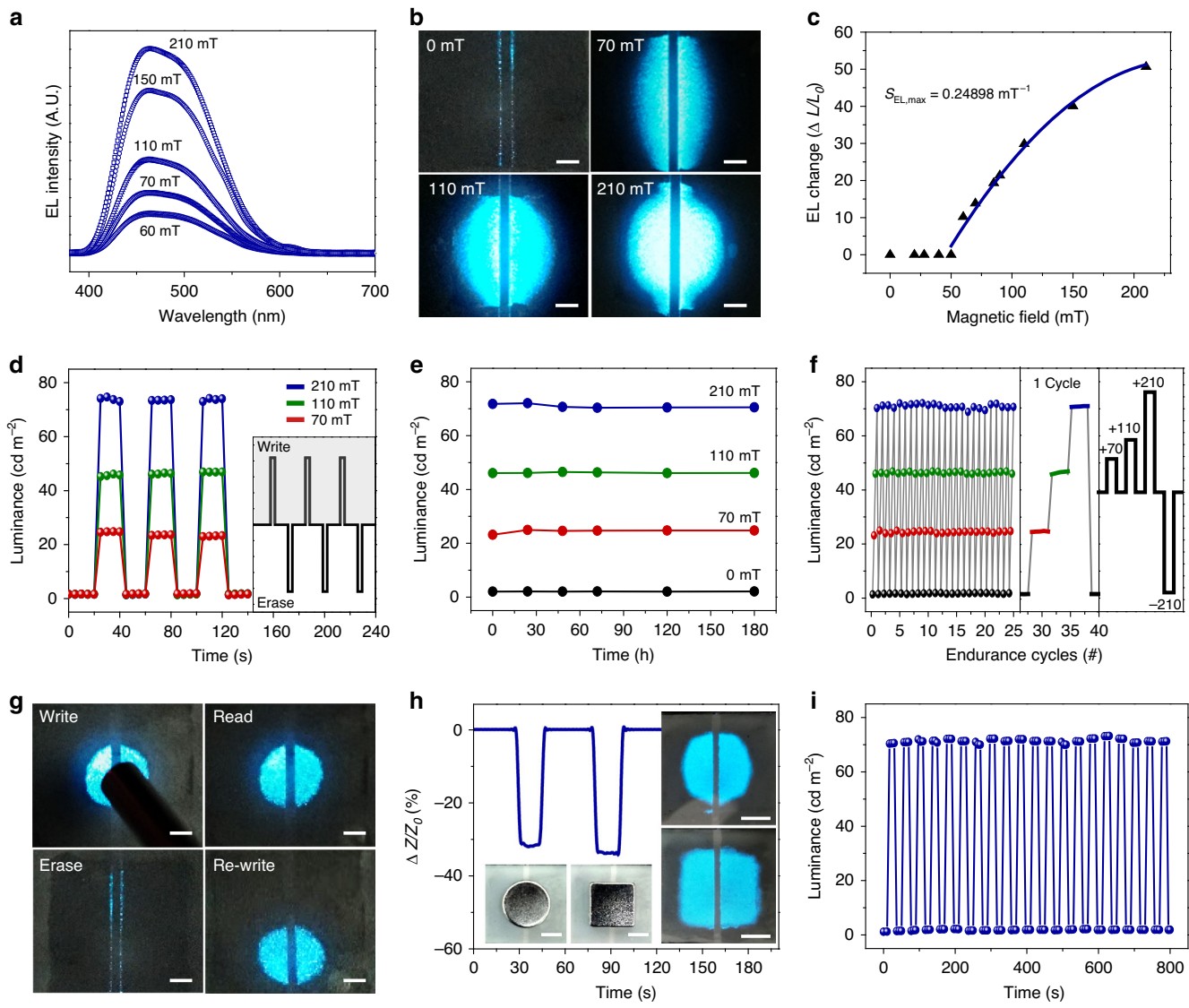

**Fig. 2 Properties of an NV-MED sensing, visualising, and memorising a magnetic field. a** EL intensity of NV-MED under different applied magnetic fields from 60 to 210 mT. **b** NV-MED under different magnetic fields, i.e. 210, 110 and 70 mT (scale bars: 1 mm). **c** Plots of the change in EL intensity of an NV-MED as a function of magnetic field. **d** Variation in EL intensity of an NV-MED in writing and erasing with different magnetic fields. A scheme of magnetic field programming for writing and erasing the EL in inset. **e** Time-dependent retention of EL intensity arising from an NV-MED with different magnetic fields. **f** Write–erase endurance cycle of EL intensity changes of an NV-MED. The inset shows the EL intensity change in one cycle and scheme of four magnetic field programmes for the endurance cycle. **g** NV-MED demonstrating the EL-based magnetic field programme/erase process (scale bars: 4 mm). **h** NV-MED with different magnetic-field shapes showing the direct visualisation of the magnetic field in EL with the device (scale bars: 5 mm). **i** Endurance cycle test of EL intensity under 210 mT magnetic field.

after the 'P' was erased, followed by rewriting letter 'L', as shown in Fig. 3g. Green emission on the arrays of the NV-MEDs was achieved by utilising the frequency-dependent colour modulation of the ZnS:Cu microparticles at ~1 kHz (Supplementary Fig. 21). The visualisation of the magnetic field in orange EL was also accomplished by simply employing the orange-emitting ZnS:Mn microparticles, as shown in Fig. 3h. Owing to the mechanical flexibility of NV-MEDs, the arrays were successfully mounted on various nonplanar surfaces of clothes and palms with reliable performance that was not significantly affected by mechanical stress, as shown in Fig. 3i.

**Patchable in vivo 3D motion tracking display.** Since the decay of magnetic field with non-contact and non-destructive characteristics is inversely proportional to the square of the distance

between two magnets, the distance between a magnetic probe and NV-MED along the z-axis can be acquired in EL, allowing for the tracking (visualisation and recording) of 3D motion of a magnetic probe buried and, thus, invisible underneath the pixelated arrays of the NV-MEDs. For demonstration, we developed a novel in vivo 3D surgery monitoring display to visualise and record the intraoperative surgery pathway, as shown in Fig. 4. The 3D surgery motion tracking display consisted of a 5 × 5 array of pixelated NV-MEDs, optical acquisition system, and magnetic surgical probe, as shown in Fig. 4a (Supplementary Fig. 22). An anesthetised rat was prepared on the operating table and 5 × 5 pixelated NV-MEDs with a total area of 2.5 × 2.5 cm$^2$ were mounted onto the labelled belly of rat and fixed with transparent medical dressing film, as shown in Fig. 4b. The magnetic surgical probe was prepared by attaching a disinfected magnet with a strength of ~200 mT to one end of the surgical probe body. Since

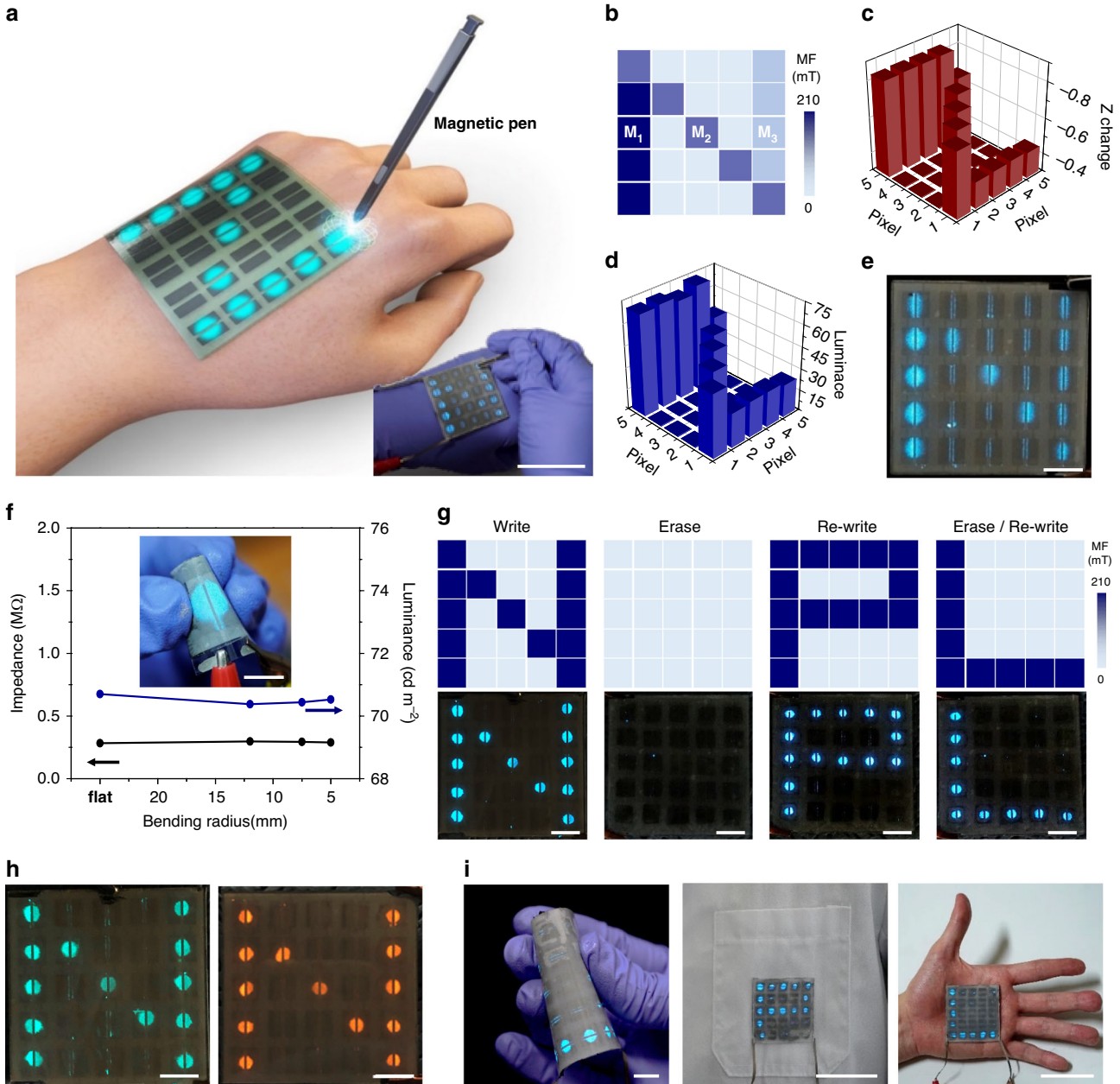

**Fig. 3 Patchable 2D NV-MED arrays. a** Schematic of a 5 × 5 NV-MED array containing parallel-type electrodes. The inset shows an image of the writing step with an NV-MED array (scale bars: 5 cm). **b** Distribution of a magnetic field programmed with different field strengths. Three different magnetic fields of 70, 110 and 210 mT were applied to develop three different marking regions of a letter 'N'. **c** Changes in impedance and **d** EL intensity in the 5 × 5 NV-MED array programmed with a different magnetic field. **e** NV-MED with the programmed magnetic field (scale bars: 1 cm). **f** Change in impedance and EL intensity of the flexible NV-MED as a function of bending radius. The inset shows an image of NV-MED during the bending test (scale bars: 1 cm). **g** Distribution of a programmed magnetic field in the writing, erasing, and rewriting steps. NV-MED array at each step are shown (bottom) (scale bars: 1 cm). **h** Photographs of green (left) and orange (right) light emission of the NV-MED array (scale bars: 1 cm). **i** Photographs of a wearable NV-MED bent by human fingers (left) and mounted on clothes (middle) and palm of a hand (right).

the arrays of NV-MEDs are mounted on rat skin, only a magnetic probe navigating the internal structure of the rat should be properly prepared for surgical implements. We confirmed that the permanent magnet was tolerant to the conventional sterilization and rarely body-absorbable. Variation of EL and impedance dependent upon the position of a magnetic probe on the channel regions of an NV-MED could occur, in particular with a magnetic probe relatively larger than a channel of a device. We confirmed that in our single NV-MED with 20 mm in width, impedance was varied and minimized at the centre of the channel

when a probe of 4 mm in diameter was scanned across the channel. Impedance variation was rarely observed when a probe was sufficiently large, compared with the channel width. No alteration in impedance occurred with a probe exactly fitted with a single NV-MED (Supplementary Fig. 23).

All the pixels of NV-MED arrays were electrically and optically pre-programmed with an AC source (voltage and frequency of ±150 V and 10 kHz) and external magnetic field (initially turned-on). As the magnetic surgical probe moves to the rat's intraperitoneal target organs, such as the liver and stomach,

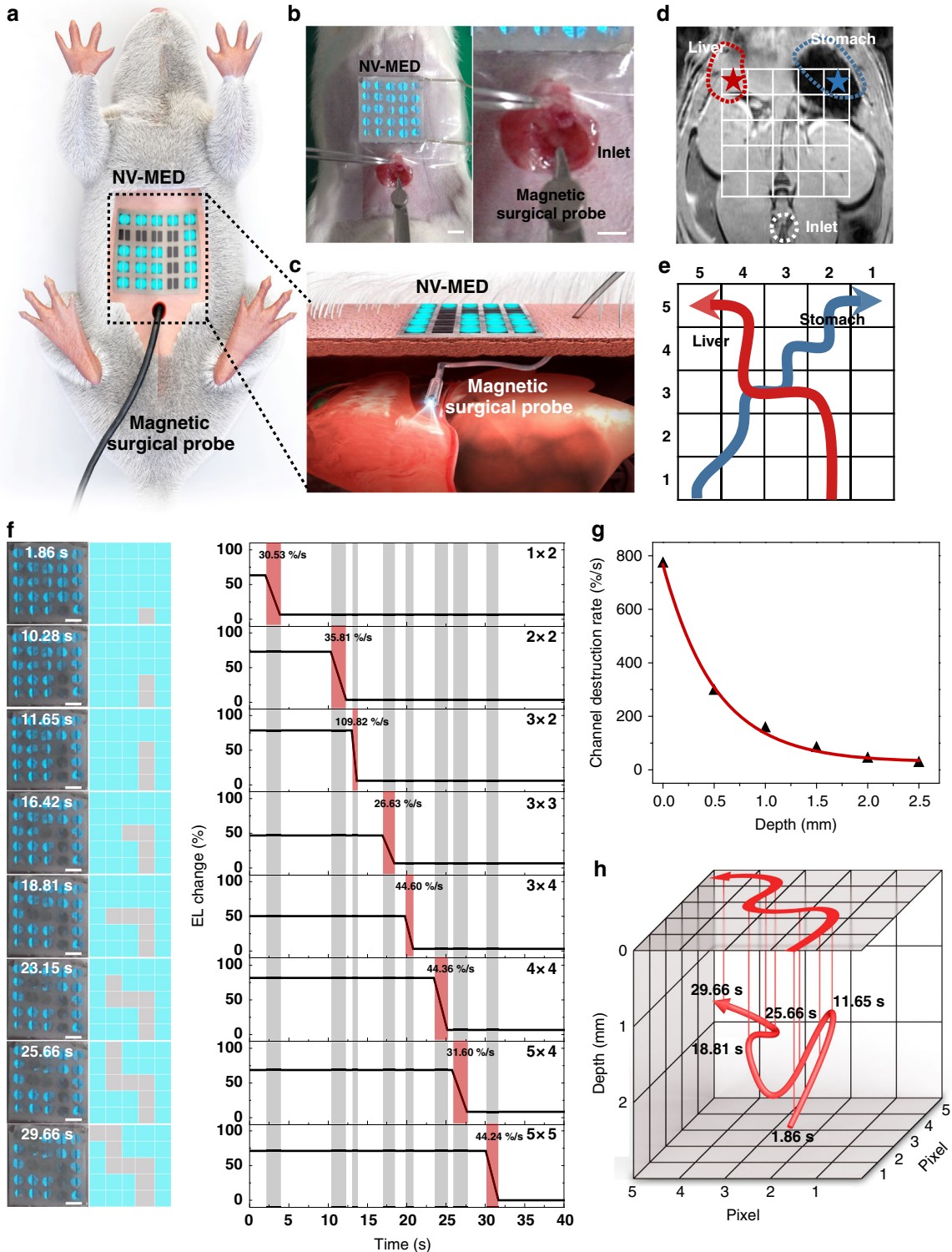

**Fig. 4 Patchable 3D motion tracking NV-MED display. a** Schematic of a 3D motion tracking display with 5 × 5 NV-MED arrays. **b** Images of NV-MED arrays mounted on the skin of a rat; the magnetic surgical probe enters the body through the inlet (scale bars: 5 mm). **c** Close-up schematic view of the NV-MED array-mounted site showing the magnetic probe below the pixels of the NV-MED arrays, resulting in the erasure of the EL of each pixel. **d** MRI scan of a rat abdominal cavity with the 5 × 5 grid. The positions of the liver and stomach were marked in red and blue, respectively. **e** The estimated routes for liver and stomach surgery with the grid. **f** EL change in the pixels of the NV-MED arrays as a function of time upon moving the magnetic probe to the target liver. NV-MED arrays mounted on a rat captured during the movement. The time for each step is shown in the photograph (scale bars: 5 mm). **g** Variation in the channel destruction rate (EL decay rate) as a function of depth of the magnetic probe. **h** 3D plots of the route to the liver with the depth of the magnetic probe showing the actual pathway of the probe.

surgical pathways can be electrically and optically stored to the display, with the characteristic EL recorded on each pixel with the precise x, y and z location, as schematically shown in Fig. 4c. An anatomical magnetic resonance imaging (MRI) scan of the abdominal organ of the anesthetised rat was performed before the intraoperative injection of the magnetic surgical probe to confirm the location of the target liver and stomach, as shown in Fig. 4d. The positions of the liver and stomach were marked in red and blue, respectively. Based on the MRI scan, the 2D x and y pathways of the probe towards the target organs were roughly set on the pixelated NV-MED with a 5 × 5 grid, as shown in Fig. 4e. All the NV-MED pixels were initially turned on, and the pixels interacting with the internal magnetic probe upon the movement to a target organ were supposed to turned off, providing a guiding EL pathway, as follows.

A series of images (consecutive frames) of the pixelated NV-MED was recorded for ~30 s, while a magnetic surgical probe passed inside the abdominal cavity towards the liver, as shown in Fig. 4f. When the probe was located at pixel (1,2), the pixel was turned off, followed by the switching-off of pixel (2,2) when the probe arrived at the pixel. The pixels (3,2), (3,3), (3,4), (4,4), (5,4), and (5,5) were sequentially turned off upon the movement of probe to the target, as shown in Fig. 4f. Next, we analysed the EL decay rate at each turned-off pixel, which was correlated with the z-distance between the probe and pixel on the skin through precise impedance analysis (Supplementary Fig. 24). We used the EL decay time which was, in turn, converted to decay rate because it was more reliable to estimate the vertical distance than luminance change. In our NV-MED, the times required to completely erase an initially programmed luminance were evaluated, dependent upon magnetic fields (Supplementary Fig. 25). The EL decay rate was evaluated as a slope of each EL decay time curve at a certain magnetic field for identifying a vertical distance. The identification of a vertical distance by the EL decay rate was much less affected by probe speed, making the 3D tracking with our NV-MED more reliable, irrespective of the probe speed. The reliable depth characterization was made in our device with a probe speed less than ~5 cm/s, which was calculated by the channel width of an NV-MED divided by the shortest EL decay time of 0.13 s under the magnetic field of 210 mT (Supplementary Fig. 26).

In our 3D motion tracking display, the channel destruction rate (EL decay rate) was inversely proportional to the z-distance with the fitted relation, as shown in Fig. 4g. Based on the EL change slope for each turned-off pixel in Fig. 4f, we constructed a 3D pathway from the abdominal cavity to the liver, as shown in Fig. 4h. Similarly, a 3D pathway to the rat's stomach was conveniently realised (Supplementary Figs. 27 and 28). Notably, owing to the non-volatility of the constructed information, the programmed pathways could be repetitively used without additional data analysis. The accuracy of the 3D pathway characterized by the arrays of NV-MED was rarely estimated because the real 3D pathway was hardly identified in in vivo experiments due to complicated physiological variations in the internal structure. In order to address this issue, we fabricated a model system with a known 3D structure by 3D printing technique. The model was attached on the inner surface of a human skull dummy and the arrays of NV-MEDs were attached on its outer surface. A 3D pathway was characterized by the method described in the manuscript and the results clearly show that the estimated 3D pathway by our NV-MEDs was almost identical with one designed by 3D printing within experimental uncertainty (Supplementary Fig. 29).

Our 3D motion tracking display enabled by the magneto-interactive EL can be used in various potential applications in biomedical engineering, such as (i) the magnetoactive micro-robot and surgical probe, which can monitor inner-body structures, (ii) low-cost surgical monitoring system as a complementary technique to the MRI and CT methods, (iii) surgical diagnosis without marking, which is required in endoscopy, and (iv) surgical recording system with feedback during and after surgery. It should be also noted that non-volatility of the programmed EL in our NV-MED is advantageous, compared with a conventional volatile sensing display (Supplementary Fig. 30). In a conventional sensor, information detected from the device should be stored in a memory before being visualized through a display. On the other hand, our NV-MED does not require a memory because the information detected by the sensory part is stored in a non-volatile manner. Although the proposed 3D motion tracking display is certainly useful for monitoring and recording invisible 3D structures buried underneath, technological limitations exist that need to be resolved from the viewpoint of the magnetic sensitivity, spatial display resolution, and operation safety.

## Discussion

In summary, we demonstrated a skin-patchable human extra-sensory interactive display enabling the simultaneous sensing, visualisation, and memorisation of a variety of magnetic fields. Non-volatile, rewritable magneto-interactive EL was developed as a function of magnetic field in an AC EL device elaborately designed with a magnetoactive conducting channel. The arrays of our magneto-interactive EL displays allow the visualisation and memorisation of a variety of information that is magnetically programmed on the arrays. Furthermore, by utilising the characteristic features of the magnetic field, which depend upon the distance along the z-axis between the magnetic probe and magneto-interactive EL display, we acquired motion information of a magnetic probe in space, thereby developing a novel 3D motion tracking display. A non-destructive surgery path guiding display was demonstrated wherein the pathway of a surgery robotic arm with a magnetic probe was visualised and recorded on our display patched on the abdominal skin of a rat, helping the arm to find an optimal pathway without the possibility of exerting damage on the way to the surgery spot.

## Methods

**Materials**. Green (D512S, ZnS:Cu) and orange (D611S, ZnS:Cu,Mn) micro-particles were purchased from Shanghai KPT Co. Poly(vinylidene fluoride-cotri-fluoroethylene-co-chlorofluoroethylene) [P(VDF-TrFE-CFE)] was purchased from Piezotech, Inc. Short MWNTs (US4365) grown by chemical vapor deposition and purified to over 95 wt% were manufactured at US Research Nanomaterials, Inc., Houston, USA. $FeCl_3 \cdot 6H_2O$ and $FeCl_2 \cdot 4H_2O$ were purchased from Sigma-Aldrich. NaOH was purchased from Daejung Co. N-hexadecane was purchased from Alfa Aesar. All other chemicals were purchased from Sigma-Aldrich and used as received. VHB 4905 and 4910 were purchased from 3 M and used as received.

**Synthesis of $Fe_3O_4$-MWNT magnetoactive fluid**. In a typical fabrication experiment, the as-treated MWNTs (100 mg) were dispersed in aqueous solution, followed by heating to 70 °C with stirring. After 10 min, $FeCl_3 \cdot 6H_2O$ was added to the suspension, followed by the addition of $FeCl_2 \cdot 4H_2O$. $NH_3 \cdot H_2O$ (30 ml, 6 wt %) was dropwise added into the suspension. After 30 min of vigorous stirring, the resulting suspension was centrifuged, and the precipitates were collected, followed by washing with deionised water and drying in an oven (80–100 °C). For the fabrication of the magnetoactive fluid, a portion of the as-prepared $Fe_3O_4$-MWNTs was dispersed in n-hexadecane by ultrasonication for 10 min[35]. Various weight ratios of $Fe_3O_4$-MWNT to hexadecane were applied to get the optimised performance.

**Fabrication of a non-volatile magneto-interactive electroluminescent display array**. First, bottom ITO electrodes with a 1 mm-sized gap were developed through sputtering with a mask onto a PET substrate, resulting in two in-plane ITO electrodes on the substrate. The thickness and sheet resistance of the ITO electrodes were 80 nm and 20 Ω cm⁻², respectively. The substrate with ITO electrodes was sequentially cleaned with acetone and 2-propanol (twice each) in an ultrasonic bath for 10 min. ZnS:Cu/PVDF-TrFE-CFE composite was prepared by mixing the ZnS:

Cu powder with a PVDF-TrFE-CFE solution with a weight ratio of 3:7. A liquid solution mixture was spin-coated on the ITO-coated substrate at 20,000 rpm for 60 s and subsequently annealed at 80 °C for 12 h. The VHB film was transferred onto the ZnS:Cu/PVDF-TrFE-CFE layer and used as a spacer for the magnetoactive fluid. The as-prepared magnetoactive fluid was poured into the VHB film spacer, followed by sealing the fluid with a PET cover.

**In vivo animal imaging and surgery**. All experimental procedures were approved by the Institutional Animal Care and Use Committee at Inchon National University. Male Sprague–Dawley rats (8 weeks old) were imaged through MRI and in vivo 3D surgery monitoring system experiments. MRI measurements were performed using Biospec 47/40 USR (Bruker, Ettlingen, Germany) with a 72-mm volume coil. During the MRI measurements, the anaesthesia of the rats was maintained using 2.0% isoflurane, and the body temperature was kept at 37 ± 1 °C using a warm bed. The animals were anesthetised with a mixture of ketamine (100 mg/kg) and xylazine (10 mg/kg), intraperitoneally for the 3D surgery monitoring system experiments. A transverse peritoneal incision of 5 mm was made with a scalpel on the middle part of the abdomen. After the peritoneal cavity was accessed, the magnetic surgical probe was inserted into the abdominal cavity for demonstration of our surgery system.

**Characterisation methods**. Transmission electron microscopy (TEM) images were performed with a JEM-F200 TEM system, coupled with the use of carbon-coated copper grids. X-ray powder diffraction (XRD) measurements were performed using HR-XRD (SmartLab). Fourier transform infrared spectroscopy (FT-IR) was conducted using Vertex 70 FT-IR system. Magnetic properties were measured by a vibrating sample magnetometer. The impedance measurement was performed with a precision LCR (inductance, capacitance, and resistance) meter (Agilent E4980A). The frequency was varied from 100 Hz to 100 kHz. The luminance and EL spectra of the devices were obtained using a spectroradiometer (Konica CS 2000). A function generator (Agilent 33220A) connected with a high-voltage amplifier (TREK 623B) was used for EL driving of the NV-MED. The device structure and the entire measurement system used to evaluate the impedance and luminance were the same ones in our previous research paper[8,11,16]. All measurements were performed under ambient conditions in air.

## Data availability
The data sets generated and analysed during this study are available from the corresponding author on reasonable request.

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

## Acknowledgements
This study was supported by the Creative Materials Discovery Program through the National Research Foundation of Korea (NRF) funded by the Ministry of Science and ICT (2018M3D1A1058536). This study was also supported by a grant from the National Research Foundation of Korea (NRF) funded by the Korean government (MEST) (No. 2020R1A2B5B0300269711).

## Author contributions
S.W.L. and S.B. conceived and designed the experiments. S.W.P., H.S., S.Y. and J.H.A. performed in vivo animal imaging and surgery. M. K., E.H.K., S.L. and W.J. performed the fabrication and demonstration on the NV-MED and prepared the figures. H.K., C.P. prepared the materials and performed device characterisation of the NV-MED. G.K. and W.S. calculated an electrical field based on finite element analysis. C.P. supervised the project, analysed the data, and wrote the paper. All authors discussed the results and commented on the paper.

## Competing interests
The authors declare no competing interests.
