## [Peer Review File · Nature Communications]

REVIEWER COMMENTS

Reviewer #1 (Remarks to the Author):

Lee et al report a magneto-interactive electroluminescent device (NV-MED) for magnetic field sensing, visualization and memorization and demonstrate its use in the surgical probe tracking application. Compared to other types of magnetic field sensors, the main advantage of NV-MED is simultaneous visualization and sensing, which is achieved from an elegant device design that incorporates the electroluminescent display.

Critical issues, however, must be clarified. Therefore, this manuscript needs major revision before considering the publication in Nature Communications Detailed issues are listed below :

- For device application, while the authors have shown the device can track magnetic field in 2D, there needs to be more evidence to show the 3D tracking ability, which is the main feature claimed in this paper.
- In the 3D motion tracking demonstration, the authors used the decay time of the magnetic field (extracted from luminance change) to characterize the distance between the magnetic probe and the NV-MED. There needs to be more evidence to show this function is valid. 1) Why is the decay time used here instead of change of luminescent intensity, which is the focus of the device characterization? If it is because the change of magnetic strength is not significant enough to cause the intensity changes, then it should be included as a limitation in the discussion. 2) Is the 3D motion tracking result (Fig. 4h) validated against the real path of the probe? 3) How does probe movement speed affect the proposed 3D tracking method? It seems like the speed of probe movement needs to be held constant for this tracking method to work.
- What is the reason behind choosing n-hexadecane as the magnetoactive fluid? It is classified to have fatal health hazard if swallowed or enter airways. And thus, it is not a good candidate for the surgical application demonstrated in this paper. Is there a biocompatible alternative?
- In the characterization sections, the authors have shown the impedance is more sensitive to magnetic field of 0-100 mT while the luminescence is not sensitive to magnetic field < 50 mT. How does AC frequency affect these sensitivities? Would it bring improvements if the frequency input is varied for different range of magnetic fields?
- Fig. 2a uses different colored lines representing the different EL intensities under various magnetic fields. A suggestion is to replace the color-line representation with legend differentiation. As colored lines in a spectrum plot lead people to think the data have different colors, while the EL emits blue light for all magnetic fields.
- Writing needs improvements. Currently some sentences are too long and arranged too strangely to get the message through, especially in the introduction section. An example is "In addition, erasing an EL by destroying the channel by applying opposite magnetic field, followed by writing a new EL with another input magnetic field, is successfully performed, making our nonvolatile magneto-interactive EL display (NV-MED) rewritable (Supplementary Fig. 1)."
- Overall, there is a lack of data in Figure 4, and further analysis and discussion are needed.
- In the application part (3D motion tracking), is the non-volatile characteristic of NV_MED necessary for the 3D motion tracking ability? If the magnetic probe continually returns to the same position, doesn't the non-volatile characteristic disturb the motion tracking? The authors should add a discussion about this issue and suggest a solution.
- According to Figure 4f, it seems that it takes a few seconds for NV-MED to react to the magnetic probe. What is the limit of speed that NV_MED can track?
- When the magnetic probe is positioned at the center of the pixel or the side, the luminance and impedance changes of the device should be different even at the same depth. Is there any way to compensate for these positional errors?
- In figures 1 and 2, the change in impedance intensity of an NV-MED shows a significant difference in the magnetic field of 0 to 50 mT, but there is no change in luminance at the same magnetic field range (0 to 50

mT). The authors should discuss this reason. Also, is the frequency of AC related to the turn-on magnetic field of the NV-MED?

- In Figure 1h, Pulse magnetic fields (70, 110 mT) were applied to the NV-MED. Why does the impedance continue to decrease after the 70, 100 mT magnetic field disappears? The author must add a discussion as to why the impedance continues to decrease after the magnetic field disappears.

Reviewer #2 (Remarks to the Author):

The authors have used a magnetic field dependent conductive gate to tune the ac electroluminescent (EL) display and to produce non-volatile and rewritable magnetic field dependent EL display. They have further demonstrated patchable devices on skin, which is capable of sensing, visualizing, and storing magnetic field information, potentially for 3D motion tracking. The experimental results are quite interesting, and the significance of the work is well articulated. I would like to recommend the authors to address the following questions before its acceptance for publication in Nature Communications.

1) Given the surface morphology and the distribution of Fe₃O₄ clusters on the MWCNT, the data shown in S-Fig. 9 are much expected. At the same time, MWCNTs can have a wide range of diameter and length. I would expect that the length of MWCNTs, in particular, will significantly affect the impedance at the given field and Fe₃O₄ concentration. Have the authors tried different MWCNTs in their experiments.

2) The statement of "...because the nanoparticles behave as a highly switchable resistor" in the sentence of "The degree of percolation of the Fe₃O₄-MWCNT network, which depends upon the external magnetic field, significantly affects the total NV-MED impedance because the nanoparticles behave as a highly switchable resistor" is not accurate.

3) It will help the readers a lot if the authors could quantitatively define the speed of the EL response. The statement of "The NV-MED exhibited a fast EL response, with its relaxation times following the square-shaped consecutive ON and OFF pattern of the magnetic field, as shown in Fig. 2d."

4) Given the much lower drift speed of Fe₂O₃ decorated MWCNT in the fluid, the device response time is probably limited by the viscosity of the magneoactive fluid.

5) The authors should comment on the safety related issue if the device (attached on skin) has to be operated at a voltage of 150V.

Reviewer #3 (Remarks to the Author):

This paper introduces a rather new type of electroluminescent device that is based on an older ACTFEL (ZNS:Cu) type of emitter. It shows a clever way to place a floating back electrode onto a transversely stimulated ACTFEL and have that back electrode only active in the presence of a magnetic field. The idea is that having the emission be sensitive to a local magnetic field, that field could then be mapped by simply looking at active vs. inactive pixels.

While the device architecture is certainly new and the application of this device is also well done (applied to a mouse model using magnetized surgical implements), the science and physical models are not. This is a rather good bit of engineering but there are a number of ways it could be done. For instance, using triplet and singlet double emitter layer and AC driving can make an AC-OLED device equally as sensitive to local fields and this has already been reported - though it wasn't carried to the extreme in demonstration as this work. And so this work does show the utility of building such devices as well as introducing a new type of such a device.

The main problem I have with the manuscript is really the writing. I emphasize that this is not an english

problem, the authors seem to have good command of the presentation. However, it is very hard to come to the main points in the manuscript due to the heavy use of jargon and otherwise irrelevant detail. The premise behind the device is quite simple, the graphics do a good job of showing how it all works, but the explanation seems to try to sell ALL the properties of this device in each sentence. It is very easy to get lost and miss the main points all together. Simplifying the language and stating mechanisms exactly would be of tremendous help in bringing out the utility the authors are trying to highlight.

As a quick side note, I might also mention that usually surgical implements are made from nonmagnetic stainless (so called surgical steel). It must be heated in autoclaves to clean. And so I am not sure if the authors can really expect the industry to change this. However, this is a small point since such sensing could be useful in other applications.

So in summary. The work builds a novel form of a device with which we are all familiar. In this form one, floating contact becomes sensitive to the local magnetic fields and this governs light production. The paper goes to great lengths to prove this and the work is carefully done. The bio applications proposed are shown to be feasible and this work too is carefully done. I recognize that there have been several papers of this type - where a modified form of an existing device is used in a biological application such as pressure sensing, that have gotten some attention lately. So this may be appropriate for NCOMM. audiences. However the form of the paper makes it difficult to read and understand. While the jargon is defined within the writing, it is so plentiful that it is hard to see the main points of the paper. Indeed, it is almost easier to understand if one were to just look at the graphics, which are well done.

Reviewer #1 (Remarks to the Author):

Lee et al report a magneto-interactive electroluminescent device (NV-MED) for magnetic field sensing, visualization and memorization and demonstrate its use in the surgical probe tracking application. Compared to other types of magnetic field sensors, the main advantage of NV-MED is simultaneous visualization and sensing, which is achieved from an elegant device design that incorporates the electroluminescent display.

Critical issues, however, must be clarified. Therefore, this manuscript needs major revision before considering the publication in Nature Communications Detailed issues are listed below:

1. For device application, while the authors have shown the device can track magnetic field in 2D, there needs to be more evidence to show the 3D tracking ability, which is the main feature claimed in this paper.

In the 3D motion tracking demonstration, the authors used the decay time of the magnetic field (extracted from luminance change) to characterize the distance between the magnetic probe and the NV-MED. There needs to be more evidence to show this function is valid. 1) Why is the decay time used here instead of change of luminescent intensity, which is the focus of the device characterization? If it is because the change of magnetic strength is not significant enough to cause the intensity changes, then it should be included as a limitation in the discussion. 2) Is the 3D motion tracking result (Fig. 4h) validated against the real path of the probe? 3) How does probe movement speed affect the proposed 3D tracking method? It seems like the speed of probe movement needs to be held constant for this tracking method to work.

Response: We appreciate the reviewer's comments to request validity of our 3D tracking display, in particular, for characterizing 3D pathways based on the information obtained from 2D arrays of the displays.

(1) As the reviewer pointed out, the luminance change could be used to characterize the distance between the magnetic probe and the NV-MED. We, however, used the EL decay time which was, in turn, converted to decay rate because it was more reliable to estimate the vertical distance than luminance change. In our NV-MED, the times required to completely erase an initially programmed luminance were evaluated, dependent upon magnetic fields, as shown in Figure S25 below. The results show that luminance change apparently depends upon the measurement time. At a given measurement time of 2 seconds, for instance, the magnetic fields of 210, 150, 110 and 70 mT exhibit all the same luminance change of almost 100 % while those of 50 and 40 mT result in the change of approximately 50 and 20 %, respectively. Although the magnetic fields are correlated with the vertical distance (depth), as shown in Table S1, the results based on the luminance change did not provide appropriate correlation of the magnetic field with the vertical distance. As the reviewer also mentioned in the following question, the identification of a vertical distance by the decay rate was much less affected by probe speed, making the 3D tracking with our NV-MED more reliable, independent to the probe speed. The decay rate which can be evaluated as a slope of each decay time curve at a certain magnetic field is more useful for identifying a vertical distance, as explicitly explained in the manuscript. Since the method is based on measuring a slope of a luminance-decay time plot at a given time, the reliable depth characterization was made in our device with a probe speed less than approximately 5 cm/sec calculated by the channel width of an NV-MED divided by the shortest decay time of 0.13 sec under the magnetic field

of 210 mT, as shown in Supporting information, S26. On the other hand, we observed that the plots of luminance changes as a function of vertical distances were varied, depending upon the probe speed, making this evaluation method much less reliable, as shown in Figure S29. We clearly mentioned the limitation of using EL change for 3D tracking characterization in the page 14 of the revised manuscript.

Supplementary Figure 25. Response time of the NV-MED in the EL change with various magnetic fields. **a.** The effective magnetic field as a function of probe depth. **b.** EL intensity decay of the NV-MED with time upon erasing with a given magnetic field. **c.** A plot of erasing time required to completely turn off EL as a function of probe depth. **d.** Table of the summary of experimental results shown in a, b and c.

(2) As the reviewer pointed out, the accuracy of the 3D path we characterized by the arrays of NV-MED was rarely estimated because the real 3D path was hardly identified in *in-vivo* experiments we performed with a rat due to complicated physiological variations in the internal structure. In order to address this issue, we fabricated a model system with a known 3D structure by 3D printing technique. The model was attached on the inner surface of a human skull dummy and the arrays of NV-MEDs were attached on its outer surface. A 3D path was characterized by the method described in the manuscript and the results in Figure S29 clearly show that the estimated 3D path by our NV-MEDs was almost identical with one designed by 3D printing within experimental uncertainty. The results validate the capability of 3D motion tracking with our NV-MEDs. We included these results in the page 15 of the revised manuscript.

Supplementary Figure 29. *In-vitro* 3D motion tracking NV-MED display. **a.** Schematic of a 3D printed tracking mould. **b.** A photograph of a 3D printed tracking mould mounted on the NV-MED. **c.** Designed depth profile of 3D printed mould recorded on 5×5 NV-MED arrays. **d.** Captured photographs of NV-MED arrays mounted on a transparent human skull dummy upon moving the magnetic probe along the route of the mould. The time for each step is shown in the photograph (scale bars: 5 cm). **e.** Erasing time of NV-MED measured from each pixel array. **f.** Erasing rate of NV-MED calculated from erasing time of each pixel array. **g.** Estimated depth profile of 3D motion tracking of magnetic probe. The results show almost similar estimated depth value to the designed depth.

(3) As the reviewer expected, a constant probe speed is practically applied for reliable 3D motion tracking based on our NV-MEDs. In addition, there must be the range of probe speed allowing for the tracking with reliable results. In our system, the highest speed employed for the measurement was approximately 5 cm/sec below which the 3D tracking was appropriately performed. The results also show that all similar plots of EL decay as a function of vertical depth were obtained, regardless of the probe speeds less than 5 cm/sec in an NV-MED, while the plots of luminance change as a function of vertical depth were not consistent with the probe speeds, making the characterization with EL decay rate more reliable. We discussed this issue in page 14 of the revised manuscript with a new Supporting information, S26.

Supplementary Figure 26. Characteristics of EL of the NV-MED with various magnetic probe speed. EL intensity changes as a function of depth with the probe speed of (a) 5 cm/s, (c) 5 mm/s, (e) 1 mm/s. EL decay rates as a function of depth with the probe speed of (b) 5 cm/s, (d) 5 mm/s, (f) 1 mm/s. A constant probe speed is practically applied for reliable 3D motion tracking based on our NV-MEDs. In our system, the highest speed employed for the measurement was approximately 5 cm/sec below which the 3D tracking was appropriately performed. All similar plots of EL decay rate as a function of vertical depth were obtained, regardless of the probe speeds, while the plots of luminance change as a function of vertical depth were not consistent with the probe speeds, making the characterization with EL decay rate more reliable.

2. What is the reason behind choosing n-hexadecane as the magnetoactive fluid? It is classified to have fatal health hazard if swallowed or enter airways. And thus, it is not a good candidate for the surgical application demonstrated in this paper. Is there a biocompatible alternative?

Response: We appreciate the insightful comments on the biocompatibility of the solvent used in our device. In our magneto-interactive display, a non-polar solvent is favorable because it provides the large impedance variation before and after development of a conductive bridge of MWNTs decorated with magnetic nanoparticles, making the sensitivity of our device maximized. We chose n-hexadecane as a non-polar solvent in addition because of its dispersion capability of MWNTs with its appropriate viscosity which also affected device switching rate. In the aspect of biocompatibility of n-hexadecane, it is noted that according to the material safety data sheet, n-hexadecane can cause skin problems when spilled. To address this issue of biocompatibility, we employed several skin compatible solvents to our NV-MEDs and the results are shown in Supporting information, S14. We found that both mineral oil and olive oil with the dielectric constants similar to that of n-hexadecane are suitable for our device. Those solvents are, however, more viscous than n-hexadecane, making the switching a little slower. We addressed this issue with a new Supporting information in page 7 of the revised manuscript.

Supplementary Figure 14. Electrical characteristics of Fe₃O₄-MWNTs in various biocompatible solvents. Non-polar solvent is more suitable for our NV-MED for higher sensitivity to magnetic field.

Solvent	Dielectric constant (ϵ_r)	Viscosity (Pa s)	Response Time (s)	Biocompatibility
n-hexadecane	2.09	0.0030041	0.89	X
Mineral Oil	2.1	2.1	1.26	Δ
Olive Oil	2.46	2.6	2.85	O
Triton X	2.2	6	10.07	X
PDMS	2.32	15.1	15.71	X
Water	78.54	0.00089	-	O

Supplementary Table 1. Characteristics of the various solvents suitable for NV-MEDs.

3. In the characterization sections, the authors have shown the impedance is more sensitive to magnetic field of 0-100 mT while the luminescence is not sensitive to magnetic field < 50 mT. How does AC frequency affect these sensitivities? Would it bring improvements if the frequency input is varied for different range of magnetic fields?

Response: As described in the manuscript, the impedance is more sensitive to magnetic field than EL in our NV-MED. In particular, under a magnetic field below 50 mT, the vertical AC field arising from networked CNT channel by the magnetic field was not sufficient to turn on EL in our device even though the device impedance was sufficiently lowered. As the reviewer pointed out, AC frequency significantly affected both the EL and impedance performance. The absolute impedance value decreases as the frequency increases since the impedance is a function of the frequency reciprocal. However, the sensitivity of impedance change to the magnetic field was rarely varied with frequency. To address the reviewer's comment on frequency dependent EL, we examined the EL characteristics with threshold magnetic field which could turn on EL as a function of frequency and the results are shown in the Supporting information, S17. Although absolute luminescence was enhanced with the frequency up to 10 kHz above which luminescence was rapidly dropped, consistent with other AC EL device, the threshold magnetic field of ~ 50 mT was rarely altered with the frequency. We discussed this issue in page 9 of the revised manuscript with a new Supporting information.

Supplementary Figure 17. Impedance and EL characteristics of an NV-MED as a functions of frequency. **a.** Impedance of NV-MED as a functions of frequency under different applied magnetic fields from 0 to 210 mT. **b.** EL intensity of NV-MED as a functions of frequency under different applied magnetic fields from 0 to 210 mT. **c.** Threshold magnetic field of NV-MED as a functions of frequency under different applied magnetic fields from 0 to 210 mT.

4. Fig. 2a uses different colored lines representing the different EL intensities under various magnetic fields. A suggestion is to replace the color-line representation with legend differentiation. As colored lines in a spectrum plot lead people to think the data have different colors, while the EL emits blue light for all magnetic fields.

Response: We appreciate the reviewer's comment to help readers understand. We replaced the colored legends of magnetic fields by symbolled ones in Figure 2a.

5. Writing needs improvements. Currently some sentences are too long and arranged too strangely to get the message through, especially in the introduction section. An example is “In addition, erasing an EL by destroying the channel by applying opposite magnetic field, followed by writing a new EL with another input magnetic field, is successfully performed, making our nonvolatile magneto-interactive EL display (NV-MED) rewritable (Supplementary Fig. 1).”

Response: We carefully modified writing by considering the reviewer’s comment. In particular, we modified the sentence the reviewer mentioned as follows. “In addition, an EL is erased by destroying the channel by applying opposite magnetic field. Subsequently, writing a new EL with another input magnetic field is successfully performed, making our nonvolatile magneto-interactive EL display (NV-MED) rewritable (Supplementary Fig. 1).” Several parts were re-written to avoid possible misleading as the reviewer suggested.

6. Overall, there is a lack of data in Figure 4, and further analysis and discussion are needed. In the application part (3D motion tracking), is the non-volatile characteristic of NV-MED necessary for the 3D motion tracking ability? If the magnetic probe continually returns to the same position, doesn’t the non-volatile characteristic disturb the motion tracking? The authors should add a discussion about this issue and suggest a solution.

Response: We appreciate the reviewer’s insightful comment. We believe that non-volatility of the programmed EL in our NV-MED is advantageous, compared with a conventional volatile sensing display. As shown in a schematic below, in a conventional sensor, information detected from the device should be stored in a memory before visualized through a display. On the other hand, our NV-MED does not require a memory because the information detected by sensory part is stored in non-volatile manner. It should be, however, admitted that non-volatile programming of a magnetic field could cause some technical problems including one the reviewer exactly pointed out. When a probe passes out a position previously programmed, the information of the position can be falsely detected. This problem can be resolved by erasing the programmed information at the positions overlaid by a probe, followed by programing new magnetic field information at the positions. We discussed this issue in page 15 of the revised manuscript with a new Supporting information, S30.

Supplementary Figure 30. Schematics of user-interactive 3D motion tracking display system. An external stimuli (magnetic) and Sensor-Signal Process-Memory-Display system must be comprehensively built. While volatile sensors can cover a part of the system, our NV-MED with non-volatile characteristics covers the entire sensing display system at a single device level.

7. According to Figure 4f, it seems that it takes a few seconds for NV-MED to react to the magnetic probe. What is the limit of speed that NV-MED can track?

Response: As answered previously, we carefully examined the probe speed limit for reliable characterization of 3D motion tracking. The results in Figure S25 and S26 show that the highest probe speed employed to our NV-MED is approximately 5 cm/sec below which reliable 3D tracking was achieved.

8. When the magnetic probe is positioned at the center of the pixel or the side, the luminance and impedance changes of the device should be different even at the same depth. Is there any way to compensate for these positional errors?

Response: We appreciate the reviewer's comment on variation of EL and impedance dependent upon the position of a magnetic probe on the channel regions of an NV-MED. As expected, the variation could occur, in particular with a magnetic probe relatively smaller than a channel of a device. We confirmed that in our single NV-MED with 20, 8, 4 mm in width, impedance was varied and minimized at the center of the channel when a probe of 4 mm in diameter was scanned across the channel, as shown in Figure S23. Impedance variation was rarely observed when a probe was sufficiently large, compared with the channel width. No alteration in impedance occurred with a probe exactly fitted with a single NV-MED. We discussed this issue in page 13 of the revised manuscript with a new Supporting information, S23.

Supplementary Figure 23. a. Photographs of NV-MED pixel arrays with various pixel widths and magnets with different diameters used in the magnetic surgical probe. Impedance characteristics according to the change of the position of the magnetic probe's (b) x and (c) y axis in the cases of the NV-MEDs with the pixels of 20, 8, 4 mm in width.

9. In figures 1 and 2, the change in impedance intensity of an NV-MED shows a significant difference in the magnetic field of 0 to 50 mT, but there is no change in luminance at the same magnetic field range (0 to 50 mT). The authors should discuss this reason. Also, is the frequency of AC related to the turn-on magnetic field of the NV-MED?

Response: We found that this comment is closely related to the previous comment 3. As described previously, the impedance is more sensitive to magnetic field than EL in our NV-MED. In particular, under a magnetic field below 50 mT, the vertical AC field arising from networked CNT channel by the magnetic field was not sufficient to turn on EL in our device even though the device impedance was sufficiently lowered. As the reviewer pointed out, AC frequency significantly affected both the EL and impedance performance. The absolute impedance value decreases as the frequency increases since the impedance is a function of the frequency reciprocal. However, the sensitivity of impedance change to the magnetic field was rarely varied with frequency. To address the reviewer's comment on frequency dependent EL, we examined the EL characteristics with threshold magnetic field which could turn on EL as a function of frequency and the results are shown in the Supporting information, S17. Although absolute luminescence was enhanced with the frequency up to 10 kHz above which luminescence was rapidly dropped, consistent with other AC EL device, the threshold magnetic field of ~ 50 mT was rarely altered with the frequency. We discussed this issue in page 9 of the revised manuscript with a new Supporting information.

Supplementary Figure 17. Impedance and EL characteristics of an NV-MED as functions of frequency. **a.** Impedance of NV-MED as a functions of frequency under different applied magnetic fields from 0 to 210 mT. **b.** EL intensity of NV-MED as a functions of frequency under different applied magnetic fields from 0 to 210 mT. **c.** Threshold magnetic field of NV-MED as a functions of frequency under different applied magnetic fields from 0 to 210 mT.

10. In Figure 1h, pulse magnetic fields (70, 110 mT) were applied to the NV-MED. Why does the impedance continue to decrease after the 70, 100 mT magnetic field disappears? The author must add a discussion as to why the impedance continues to decrease after the magnetic field disappears.

Response: Impedance was rapidly decreased once a magnetic field was applied to an NV-MED. Further slight decrease in impedance occurred during the magnetic pulse (2 sec). When the magnetic field was removed, the conductive CNT channels with magnetic nanoparticles were slightly relaxed in viscous solvent medium and as a result the conductivity of the channel slightly decreased, giving rise to the slight increase of impedance, as observed

in Figure 4h. When a high magnetic field was applied (e.g. 110 mT), the impedance was rarely increased after the removal of the magnetic field because the magnetic field was sufficiently large for fixating the conductive channel. We discussed this issue in page 7 of the revised manuscript with a modified Figure 1h.

Reviewer #2 (Remarks to the Author):

The authors have used a magnetic field dependent conductive gate to tune the ac electroluminescent (EL) display and to produce non-volatile and rewritable magnetic field dependent EL display. They have further demonstrated patchable devices on skin, which is capable of sensing, visualizing, and storing magnetic field information, potentially for 3D motion tracking. The experimental results are quite interesting, and the significance of the work is well articulated. I would like to recommend the authors to address the following questions before its acceptance for publication in Nature Communications.

The detailed comments are listed as following:

1. Given the surface morphology and the distribution of Fe_3O_4 clusters on the MWCNT, the data shown in S-Fig. 9 are much expected. At the same time, MWCNTs can have a wide range of diameter and length. I would expect that the length of MWCNTs, in particular, will significantly affect the impedance at the given field and Fe_3O_4 concentration. Have the authors tried different MWCNTs in their experiments?

Response: We appreciate the reviewer's insightful comment. We agree with the reviewer that MWNTs with different length and diameter will affect the impedance change of our NV-MED at the given Fe_3O_4 concentration. As the reviewer suggested, we explicitly examined how impedance of our NV-MED was varied with MWNTs having different diameters and lengths and the results are shown in Supporting information, S12a. In a given length of MWNTs, the sensitivity of impedance variation was rarely affected with the diameter of the nanotubes. As expected, the absolute impedance in a given magnetic field was much lower with long MWNTs than short ones due to facile formation of networked conductive channel with long nanotubes. However, the long MWNTs exhibited relatively low sensitivity of the impedance to the magnetic field, as shown in Figure S12b, making them less suitable for our NV-MED. We addressed this issue in page 6 of the revised manuscript with a new Supporting information, S12.

Supplementary Figure 12. Electrical characteristics of Fe_3O_4 -MWNTs with various lengths and diameters of the MWNTs. **a.** Table of impedance characteristics with the MWNTs. **b.** Changes in impedance of the MWNTs in hexadecane as a function of the magnetic field. The diameter of the MWNTs does not significantly affect the impedance characteristics. As the length of the MWNTs increases, the impedance are decreased. The magnetoactive fluid with long MWNTs are less suitable for our NV-MED due to low EL sensitivity and poor fluid stability.

2. The statement of "...because the nanoparticles behave as a highly switchable resistor" in the sentence of "The degree of percolation of the Fe₃O₄-MWNT network, which depends upon the external magnetic field, significantly affects the total NV-MED impedance because the nanoparticles behave as a highly switchable resistor" is not accurate. (page 6 line 8)

Response: We are regretful for the ambiguous sentence which might mislead the readers. We modified it accordingly. "The degree of percolation network of the Fe₃O₄-MWNTs, which depends upon the external magnetic field, significantly affects the total NV-MED impedance because the Fe₃O₄-MWNTs behave as a highly switchable conductor."

3. It will help the readers a lot if the authors could quantitatively define the speed of the EL response. The statement of "The NV-MED exhibited a fast EL response, with its relaxation times following the square-shaped consecutive ON and OFF pattern of the magnetic field, as shown in Fig. 2d.

Response: As the reviewer suggested, we quantified the EL response as a function of magnetic field and the results are shown in Supporting information, S25 and S26. The EL decaying time results dependent upon magnetic field were successfully utilized for 3D depth tracking, as explicitly shown in Figure S29. A new Supporting table and figure were added with the quantified information in page 14 of the revised manuscript, as follow.

"We used the EL decay time which was, in turn, converted to decay rate because it was more reliable to estimate the vertical distance than luminance change. In our NV-MED, the times required to completely erase an initially programmed luminance were evaluated, dependent upon magnetic fields (Supplementary Fig. 25). The decay rate was evaluated as a slope of each decay time curve at a certain magnetic field for identifying a vertical distance. The identification of a vertical distance by the decay rate was much less affected by probe speed, making the 3D tracking with our NV-MED more reliable, irrespective of the probe speed. The reliable depth characterization was made in our device with a probe speed less than approximately 5 cm/sec calculated by the channel width of an NV-MED divided by the shortest decay time of 0.13 sec under the magnetic field of 210 mT (Supplementary Fig. 26)."

Supplementary Figure 25. Response time of the NV-MED in the EL change with various magnetic fields. **a.** The effective magnetic field as a function of probe depth. **b.** EL intensity decay of the NV-MED with time upon erasing with a given magnetic field. **c.** A plot of erasing time required to completely turn off EL as a function of probe depth. **d.** Table of the summary of experimental results shown in a, b and c.

4. Given the much lower drift speed of Fe₂O₃ decorated MWCNT in the fluid, the device response time is probably limited by the viscosity of the magnetoactive fluid.

Response: We appreciate the reviewer's insightful comment on the effect of medium viscosity on device response time. The average magnetophoretic velocity was calculated by follows:

$$v = \frac{1}{3\eta D_H} \pi R^2 h \frac{\chi}{\mu_0} \nabla |B|^2.$$

The response rate of an NV-MED decreases with the medium viscosity. To verify it, we employed several solvents with different viscosities at room temperature to our NV-MEDs and estimated how fast the devices responded, dependent upon the solvents. The results clearly show that the response time increases with solvent viscosity. We addressed this issue in page 7 of the revised manuscript with a new Supporting information, S13.

Supplementary Figure 13. Electrical characteristics of Fe_3O_4 -MWNTs in solvents with different viscosities. Changes in impedance of Fe_3O_4 -MWNTs with various solvents of (a) n-hexadecane, (b) mineral oil, (c) olive oil, (d) triton X, (e) PDMS under magnetic field (210 mT). The response time of the NV-MED increased with the viscosity of a solvent.

Solvent	Dielectric constant (ϵ_r)	Viscosity (Pa s)	Response Time (s)	Biocompatibility
n-hexadecane	2.09	0.0030041	0.89	X
Mineral Oil	2.1	2.1	1.26	Δ
Olive Oil	2.46	2.6	2.85	O
Triton X	2.2	6	10.07	X
PDMS	2.32	15.1	15.71	X
Water	78.54	0.00089	-	O

Supplementary Table 1. Characteristics of the various solvents suitable for NV-MEDs.

5. The authors should comment on the safety related issue if the device (attached on skin) has to be operated at a voltage of 150V.

Response: We appreciate the reviewer's insightful comment. As the reviewer is aware, most of EL devices based on AC suffer from high voltage operation because they rely upon field induced light emission. Tremendous efforts have been made to reduce the operation voltage of an AC EL. As the reviewer is concerned, the safety of our NV-MED is one of the most critical issues for further development, in particular for skin patchable applications. To reduce the operation voltage of an NV-MED, we employed a high-k dielectric of PVDF-TrFE-CFE. It should be noted that although the operation voltage of 150 V of our NV-MED is quite high, compared with conventional LEDs, it is significantly lower than that prepared

with conventional elastomer. Since light emitting phosphor particles are completely embedded in dielectric medium, the current during device operation is very low, making our device safely skin-patchable in spite of the rather high operation voltage. We addressed this issue in page 10 of the revised manuscript with a new Supporting information, S19.

Supplementary Figure 19. L–V characteristics of NV-MED devices with different magnetic fields. **a.** L–V characteristics of NV-MED with ZnS:Cu/PVDF-TrFE-CFE composite layer under different magnetic fields. **b.** L–V characteristics of NV-MED with ZnS:Cu/poly(dimethyl siloxane) (PDMS) composite layer under different magnetic fields. **c.** Variation in EL intensity of an NV-MED with ZnS:Cu/PVDF-TrFE-CFE composite layer under different magnetic fields. **d.** Variation in EL intensity of an NV-MED with ZnS:Cu/PDMS composite layer under different magnetic fields. The NV-MED with ZnS:Cu/PVDF-TrFE-CFE operated at the voltage lower than that with PDMS owing to the dielectric constant of PVDF-TrFE-CFE higher than that of PDMS.

Reviewer #3 (Remarks to the Author):

This paper introduces a rather new type of electroluminescent device that is based on an older ACTFEL (ZnS:Cu) type of emitter. It shows a clever way to place a floating back electrode onto a transversely stimulated ACTFEL and have that back electrode only active in the presence of a magnetic field. The idea is that having the emission be sensitive to a local magnetic field, that field could then be mapped by simply looking at active vs. inactive pixels.

While the device architecture is certainly new and the application of this device is also well done (applied to a mouse model using magnetized surgical implements), the science and physical models are not. This is a rather good bit of engineering but there are a number of ways it could be done. For instance, using triplet and singlet double emitter layer and AC driving can make an AC-OLED device equally as sensitive to local fields and this has already been reported - though it wasn't carried to the extreme in demonstration as this work. And so this work does show the utility of building such devices as well as introducing a new type of such a device.

Response: We would like to thank the reviewer for his or her positive opinion on our magneto-interactive display enabling 3D motion tracking. As the reviewer pointed out, the novelty of our work is to develop a novel AC EL architecture with magneto-interactive floating gate where magnetic field is detected, memorized and more importantly visualized. We agree that a similar magnetic field sensing could be done with other scientific principles including ones the reviewer mentioned based on magnetic field effects on the EL of organic light emitting diodes where the field modulated the ratio of the singlet/triplet exciton yield. However, visualization of a magnetic field has been rarely made in the device level as shown in our current work. We explicitly addressed this issue with the proper references (1. Masaki T., Ryo N., Hajime N. & Chihaya A. Understanding degradation of organic light-emitting diodes from magnetic field effects. *Communications Materials* 1, 18 (2020). 2. Kalinowski, J., Cocchi, M., Virgili, D., Di Marco, P. & Fattori, V. Magnetic field effects on emission and current in Alq₃-based electroluminescent diodes. *Chem. Phys. Lett.* 380, 710–715 (2003). 3. Peng, Q., Li, X. & Li, F. Time-resolved spin-dependent processes in magnetic field effects in organic semiconductors. *J. Appl. Phys.* 112, 114512 (2012).)

The main problem I have with the manuscript is really the writing. I emphasize that this is not an english problem, the authors seem to have good command of the presentation. However, it is very hard to come to the main points in the manuscript due to the heavy use of jargon and otherwise irrelevant detail. The premise behind the device is quite simple, the graphics do a good job of showing how it all works, but the explanation seems to try to sell ALL the properties of this device in each sentence. It is very easy to get lost and miss the main points all together. Simplifying the language and stating mechanisms exactly would be of tremendous help in bringing out the utility the authors are trying to highlight.

Response: We are very regretful that the details we elaborately described to explain what exactly happened in our NV-MED made the reviewer as well as potentially readers confused. Although we tried our best to demonstrate all the details of our system for helping the readers appropriately understand it, we admit our mistake we realized from the reviewer's comments. As the reviewer suggested, we carefully modified our manuscript by simplifying the language and removing some irrelevant details such as principles of dielectrophoretics vs. magneto-phoretics which, we believe, should be addressed as a separate topic following the work.

As a quick side note, I might also mention that usually surgical implements are made from nonmagnetic stainless (so called surgical steel). It must be heated in autoclaves to clean. And so I am not sure if the authors can really expect the industry to change this. However, this is a small point since such sensing could be useful in other applications.

Response: We appreciate the reviewer's comment on the suitability of our magnetic probe for surgical applications. Since the arrays of NV-MEDs are mounted on rat skin, only a magnetic probe navigating the internal structure of the rat should be properly prepared for surgical implements. In our experiments, we utilized a stainless steel bar with a permanent magnet of approximately 4 mm in width attached. We confirmed that the permanent magnet was tolerant to the conventional sterilization and rarely body-absorbable. We briefly mentioned this issue in page 13 of the revised manuscript.

So in summary. The work builds a novel form of a device with which we are all familiar. In this form one, floating contact becomes sensitive to the local magnetic fields and this governs light production. The paper goes to great lengths to prove this and the work is carefully done. The bio applications proposed are shown to be feasible and this work too is carefully done. I recognize that there have been several papers of this type - where a modified form of an existing device is used in a biological application such as pressure sensing, that have gotten some attention lately. So this may be appropriate for NCOMM. audiences. However, the form of the paper makes it difficult to read and understand. While the jargon is defined within the writing, it is so plentiful that is hard to see the main points of the paper. Indeed, it is almost easier to understand if one were to just look at the graphics, which are well done.

Response: As the reviewer suggested, we carefully modified the manuscript by reducing unnecessary languages and jargons, making it more concise and readable.

REVIEWERS' COMMENTS

Reviewer #1 (Remarks to the Author):

The paper is relevant and technically interesting and detailed. The authors have satisfactorily addressed the reviewer's prior comments and the manuscript should be published without further revisions .

Reviewer #2 (Remarks to the Author):

The authors have seriously considered the suggestions/comments and made the necessary changes to clarify the questions. Some added results and discussions are especially helpful for readers to understand the experimental results.

Reviewer #3 (Remarks to the Author):

This is the revised review of this paper after my initial comments have been addressed. After reading through the new manuscript along with the support. info. and the authors comments, I am now comfortable that my earlier concerns have been adequately met and that this new presentation is more suitable for publication. There are one or two areas of new writing (shown in red in the manuscripts) that should be addressed for english/grammar, but otherwise the presentation is clear concise and makes its point well. The study does support the overall hypothesized mechanisms of operations for this new type of imaging device. The support. Info. does further support and validate operational parameters, which are necessary to understand the limits of utility.